# Lost in translation: how can education about dementia be effectively integrated into medical school contexts? A realist synthesis

Ellen Tullo,[1] Luisa Wakeling [ID],[2] Rachel Pearse,[3] Tien Kheng Khoo,[4,5] Andrew Teodorczuk[5,6,7,8]

For numbered affiliations see end of article.

**Correspondence to**
Dr Luisa Wakeling;
luisa.wakeling@ncl.ac.uk

## ABSTRACT

**Objectives** The prevalence of dementia in both community and hospital settings requires a clinical workforce that is skilled in diagnosis and management of the condition to competently care for patients. Though evidence of successful educational interventions about dementia exists, effective translation into medical school curricula is the exception rather than the norm.

**Design** We adopted a realist synthesis approach following Realist And MEta-narrative Evidence Syntheses: Evolving Standards (RAMESES) guidelines to answer the following questions: (1) what are the barriers to integrating effective interventions about dementia into medical school curricula and (2) where they are successfully delivered, what are the contextual factors that allow for this enactment?

**Data sources** We searched PubMed, Embase, CINAHL and PsycINFO using the MesH terms *Schools, Medical; Students, Medical; Education, Medical* AND *Neurocognitive disorders* or the closest possible set of terms within each database.

**Eligibility criteria** Undergraduate or graduate entry medical school programme, teaching and learning focussing on dementia, evaluating student outcomes (satisfaction, knowledge, skills, attitudes or behaviours), interventions described clearly enough to classify teaching method, any research design (quantitative and qualitative), English language.

**Data extraction and synthesis** We used a shared spreadsheet to enter key information about eligible studies and the reasons for excluding studies that did not fit eligibility criteria. We extracted descriptive data about the nature of educational interventions and narrative information as to barriers and facilitators to implementing those interventions.

**Results** Our initial literature search identified 16 relevant papers for review. Systematic extraction of data informed the development of an initial programme theory (IPT) structured around four contextual barriers: 'culture', 'concern for patient welfare', 'student attitudes' and 'logistics' with associated facilitatory mechanisms embed medical education about dementia.

**Conclusions** We outline the process of generating our IPT, including overlap with Cultural Historical Activity Theory. We outline our intention to refine our programme theory through ongoing review of the evidence base and collaboration with stakeholders, with the aim of

## STRENGTHS AND LIMITATIONS OF THIS STUDY

⇒ Realist reviews allow flexibility to refine study methodology in parallel with literature review in order to explain how complex educational interventions might work (or not) in particular contexts.

⇒ A realist review deciphers what works for whom, in what circumstances, offering a more pragmatic answer for educators interested in improving medical education about dementia in their own institutions than a traditional systematic review.

⇒ Our study sets out an initial programme theory (IPT) that can be used as a tool to help educators integrate effective education about dementia into the undergraduate curriculum internationally.

⇒ Our IPT has not yet been reviewed and refined by stakeholders outside of our study team.

⇒ The longitudinal viability of integrating dementia education into the undergraduate curriculum remains unanswered by this study.

finalising a model for successful integration of dementia education.

## INTRODUCTION

With the ageing population, dementia is an increasingly prevalent illness. Prevalence increases with age such that one in six people in the UK aged over 80 have dementia.[1] Moreover, there is evidence internationally that care for patients with dementia is poor, and staff feel unequipped to manage people with dementia (PWD).[2 3] In part, this can be attributed to gaps not only in skills and knowledge,[4] but also in attitudes.[5 6] Attitudes are shaped by early education experiences and doctors as leaders in healthcare have an important role in setting the compass of care.[7]

Unfortunately, education about dementia in medical schools remains patchy.[8] There is piecemeal evidence from international studies that discrete interventions can

improve students' knowledge of and attitudes towards PWD. Even brief interventions that incorporate elements of experiential learning such as simulation[9–11]; and story-telling[12] have been shown to have at least a short-term positive impact on student knowledge and attitudes, although longer-term impacts and sustainability remain unclear. This may be because of the fact that healthcare settings are not set up to facilitate good dementia practice. Care home placements and clinical clerkships facilitating contact with PWD likewise appear to improve knowledge and attitudes,[13 14] but are infrequently implemented.

There are also examples of more extensive educational interventions in the form of longitudinal pairings of medical students with PWD. In the USA, the Partnering in Alzheimer's Instruction Research Study (PAIRS) Program[15] developed from an original buddy programme[16] brings together medical students and PWD over a period of 1 year.

Mixed methods evaluation suggests enhanced knowledge, skills and positive attitudes among medical students who participated, although students were self-selected and therefore likely to have held predisposing positive attitudes.

The PAIRS model has been adapted in the form of the 2-year *Time for Dementia* programme, run collaboratively between a medical school and the UK Alzheimer's Society.[17] Evaluation of *Time for Dementia* is ongoing, but there is evidence from a controlled trial that the intervention led to improvements in medical students' knowledge and attitudes[18] and also has potential benefits for the patients and families.[19]

Longitudinal programmes involving PWD however are the exception rather than the norm—medical school education about dementia remains inadequate to sufficiently equip graduates to meet the needs of PWD. Given the imperative to improve care for PWD and the existence of evidence as to effective practice, it remains unclear why educational interventions about dementia are not implemented more widely across medical schools. Arguably, this is because curricula are predicated on rigid historically structured content; to bring in change would involve challenging the traditional status quo as 'ologies' risk being replaced and thereby devalued. Moreover, systems of care may be slow to adopt good dementia practice and so the hidden curriculum limits the ability to innovate and perceived importance of good dementia education initiatives.

## Rationale and research intent
Our initial intention was to review existing educational interventions about dementia to identify which features of such interventions led to enhancement in medical students' knowledge, skills, attitudes and behaviours. However, our initial review of the literature, including studies cited above, suggested sufficient evidence of a range of effective models of teaching and learning about dementia. Aligned to the research intent, the more pertinent questions are, first, why effective interventions are

not more frequently and systematically integrated into medical school curricula? Second, for whom and in which circumstances is successful dementia education effectively implemented in medical school curricula? We reasoned that this question might be better answered using a realist synthesis rather than a more traditional systematic review.[20] Such an approach aims to decipher what works for whom, in what circumstances and is therefore likely to provide a more helpful answer for educators interested in improving medical education about dementia in their own institutions.

## METHODS
### Changes to the review process
As outlined above, the focus and purpose of our review has already evolved. An accepted feature of realist synthesis is that purpose, scope, inclusion criteria and method of data synthesis are refined in an iterative manner as the review progresses.[21] The details of the evolution of our study purpose are included in online supplemental appendix 1. Our report is structured according to Realist And MEta-narrative Evidence Syntheses: Evolving Standards (RAMESES) publication standards.[22]

### Rationale for realist approach
We have suggested that there is already sufficient evidence as to effective methods for delivering medical education about dementia and an implementation gap. Therefore, a traditional systematic review to highlight key features of effective practice is unlikely to answer the question as to why education about dementia remains under-represented in medical curricula or align to the research intent. Rather, there is a need to improve understanding of the barriers to implementation and the facilitators in institutions that have successfully introduced effective practice.

Realist reviews add explanation as to how interventions might work (or not) in particular contexts to produce change.[23] The determinant of a desired outcome, in this case improving medical students' ability to care for PWD, is unlikely to be simple or linear—rather, multiple mechanisms are likely to be relevant.[21] These underlying mechanisms may not be intrinsic to the educational intervention itself, but dependent on the historical context, location and the participants involved. For example, in the case of dementia, public and professional attitudes (sometimes ageist) have changed significantly over the last decades,[5] and this may have a bearing on the gap between the expanding needs of PWD and its proportional under-representation in the medical curriculum. A traditional systematic review is unlikely to capture this contextual issue.

A realist approach has been used to examine the complex underpinnings of a range of polemics in medical education including supporting international medical graduates into the UK workforce,[24] the supervision and appraisal process of doctors,[25 26] and how healthcare

training translates into benefits for patients.[27] A realist synthesis will similarly allow us to develop a 'programme theory'—a diagram to lay out the contexts, mechanisms and outcomes in relation to the curricular integration of undergraduate (UG) education about dementia.

## Data sources

We drafted the following eligibility criteria before conducting an exploratory search of databases in (PubMed, Embase, CINAHL, PsycINFO) using the MesH terms *Schools, Medical; Students, Medical; Education, Medical* AND *Neurocognitive disorders* or the closest possible set of terms within each database (see online supplemental file for details of the literature searches).

## Eligibility criteria

► Undergraduate or graduate entry medical school programme.
► Teaching and learning focussing on dementia (rather than delirium or cognitive impairment more broadly).
► Evaluating student outcomes>satisfaction, knowledge, skills, attitudes or behaviours.
► Interventions described clearly enough to classify teaching method.
► Any research design, including quantitative and qualitative.
► English language.

## Data extraction and synthesis

We divided the articles between our research team and constructed a shared spreadsheet to enter key information about eligible studies and to keep track of the reason for excluding studies that did not fit eligibility criteria. We extracted descriptive data about the nature of educational interventions and narrative information as to barriers and facilitators to implementing those interventions. We used the template as a basis for discussion of key findings at regular team meetings and to note any concerns about the quality of the included studies.

We derived our initial programme theory (IPT) from synthesis of the key barriers and facilitators to implementation emerging from the initial literature review and data extraction as described in the results. This is the first step in our process of realist synthesis—proposed ongoing realist methodology to iteratively test and refine the model is outlined in the discussion.

## Patient and public involvement

None at this stage.

## RESULTS
## Literature search

Our initial database searches yielded 358 papers. In total, 23 were duplicates, 238 were excluded by title and 4 were not in English. We reviewed 93 papers by abstract or full text. A total of 77 papers did not meet our eligibility criteria, leaving 16 studies for data extraction (figure 1).

Manual searching of the reference lists of the 16 eligible studies, and our team's awareness of pertinent literature beyond the remit of the search highlighted that our initial database search terms had not been sufficiently broad to identify all relevant interventions. This may be due to medical education research being referenced across different database types (biomedical, psychiatric, educational) that each use a range of search terms in relation to dementia, medical education and the prequalification stage of medical training. We included a further four relevant studies in our initial synthesis to inform our IPT (table 1).

## Document characteristics

Of the 20 papers evaluating UG dementia education, 13 involved only medical students and 5 involved other healthcare professions, including pharmacy, physician assistants and nurses. One study involved staff contributing to an educational programme. Cohort sizes ranged from 12 to 386 and the lengths of the educational interventions ranged from 1 hour to 2 years. The 20 studies considered a range of outcomes including student knowledge (n=7), student attitudes (n=9) and qualitative data collected from focus groups, interviews and student reflections (n=9).

## Initial findings

Extraction of narrative data about barriers to and facilitators of implementation of each intervention revealed a number of common themes. Many studies initially highlighted the barriers towards implementing prior educational interventions about dementia as a rationale for the development of their own.

The evolution of medical education is influenced by the historical and cultural context in which it takes place. There has been a strong reliance on Flexner models of curricula that are over 100 years old—it is surprising how slowly curricular structures and content have evolved.[28] An emphasis on system-based specialties and academic disciplines means that conditions such as dementia do not 'belong' to a particular specialty and therefore may not have systematically appeared in the UG medical curriculum. This has been highlighted by national guidelines in the UK and elsewhere, leading to mandated dementia education, cited by some authors as a prompt for their institution to introduce this. Despite such a mandate, authors note persistent negative attitudes towards dementia among faculty which act as a barrier to implementation. The presence of a committed leader and faculty engagement appears to be an important facilitator to integration of dementia education.

Authors cited understandable concerns about the welfare of PWD who might have limited capacity to consent to participation in teaching, and on the potential burden to PWD and their carers. Teams that worked in partnership with PWD, carers and voluntary sector organisations in the design and delivery of dementia education were able to navigate some of these risks. Other studies

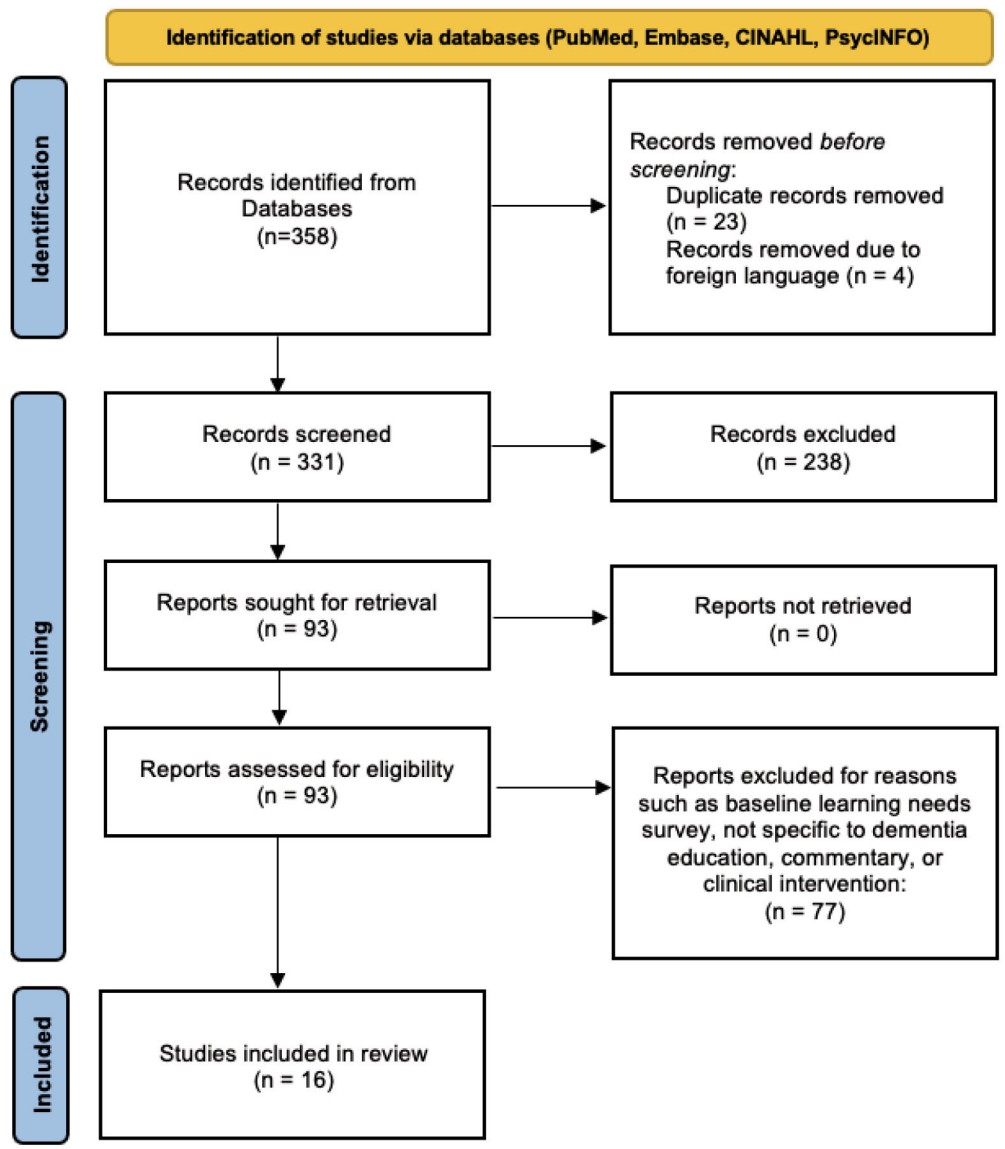

**Figure 1** Preferred Reporting Items for Systematic Reviews and Meta-Analyses flow diagram for study selection process.[33]

elected to use virtual or simulated alternatives rather than direct contact with PWD and their carers.

The attitudes of medical students were also of relevance—there were examples of negative attitudes towards dementia as a topic, with student perception that community-based teaching about dementia detracted from the more pertinent environment of acute care. As such, students were less motivated to learn about chronic conditions such as dementia in community placements than they were about acute presentations of illnesses in emergency environments. This may be associated with institutional culture that devalues care of older adults, with the negative attitudes of faculty influencing medical students, sometimes termed the 'hidden curriculum'. Despite negative attitudes, successful programmes of education about dementia instilled motivation in students through mechanisms such as sustained supervision from teachers and peers, and through systematic reflective practice.

Authors also cited the substantial logistical challenges to integration of dementia education, most notably longitudinal programmes pairing medical students with PWD. A full timetable and resource constraints challenged curriculum designers, whereas dedicated resources, administrative support, clear roles for faculty members and iterative implementation acted as facilitators to integrate dementia education.

One paper specifically explored the barriers and facilitators to implementing a longitudinal dementia education programme in a single site.[29] Their findings overlap substantially with the themes appearing across other studies, affording some validity to our initial explanation of contexts, mechanisms and outcomes in the following IPT.

### Initial programme theory (IPT)

An IPT (also known as draft or rough programme theory) '*refers to the initial sketch of a theory that is used to guide a realist*

**Table 1** Papers included in exploratory literature review

| Author | Year | Title | Participants | Outcome measure |
|---|---|---|---|---|
| Annear et al[13] | 2016 | Encountering aged care: a mixed methods investigation of medical students' clinical placement experiences | 61 Y5 MS | Knowledge (DKAT) Qualitative (FG) |
| **Banerjee et al[17]** | 2017 | How do we enhance undergraduate healthcare education in dementia? | n/a | n/a |
| **Banerjee et al[18]** | 2021 | A comparative study of the effect of the Time for Dementia programme on medical students | 386 Y2/3 MS | Knowledge (ADKS, DKQ) Attitudes (ADQ, DAS, MCRS, JSE) |
| Blazek et al[34] | 2015 | Medical students in a dementia care facility: an enhanced geriatric psychiatry experience | 99 Y3 MS | Knowledge (Likert) Attitudes (Likert) Qualitative (FG) |
| de Abreu et al[9] | 2017 | A simulation exercise to raise learners' awareness of the physical and cognitive changes in older adults | 49 students (MS, physician assistant, pharmacy, psychiatry residents) | Attitudes (ADQ) |
| **Feeney et al[29]** | 2021 | Barriers and facilitators to implementing a longitudinal dementia education programme into undergraduate healthcare curricula: a qualitative study | 12 staff | Qualitative (interviews) |
| George et al[35] | 2011 | Impact of participation in TimeSlips, a creative group-based storytelling programme, on medical student attitudes toward persons with dementia: a qualitative study | 15 Y4 MS | Qualitative (questionnaires) |
| George et al[36] | 2012 | Evaluating an arts-based intervention to improve medical student attitudes toward persons with dementia using the dementia attitudes scale | 22 Y4 MS | Attitudes (DAS) |
| George et al[37] | 2013 | An arts-based intervention at a nursing home to improve medical students' attitudes toward persons with dementia | 22 Y4 MS | Attitudes (DAS) |
| George et al[12] | 2014 | How a creative storytelling intervention can improve medical student attitude towards persons with dementia: a mixed methods study | 22 Y4 MS | Attitudes (DAS) Qualitative (FG) |
| Gilmartin-Thomas et al[38] | 2018 | Impact of a virtual dementia experience on medical and pharmacy students' knowledge and attitudes toward people with dementia: a controlled study | 278 Y3 MS and Y4 pharmacy students | Attitudes (DAS) |
| **Gilmartin-Thomas et al[10]** | 2020 | Qualitative evaluation of how a virtual dementia experience impacts medical and pharmacy students' self-reported knowledge and attitudes towards people with dementia | 53 Y3 MS and Y4 pharmacy students | Qualitative (FG) |
| Goldman and Trommer[39] | 2019 | A qualitative study of the impact of a dementia experiential learning project on premedical students: a friend for Rachel | 95 pre-MS | Qualitative (reflections) |
| Griffiths et al[40] | 2020 | Perceptions and attitudes towards dementia among university students in Malaysia | 97 MS and pharmacy students (Y unknown) | Attitudes (AADS, ATOP, IPQ) |
| Jefferson et al[15] | 2012 | Medical student education program in Alzheimer's disease: the PAIRS Program | 45 Y1 MS | Knowledge (BPDKT, BUPPDKT) Qualitative (reflections) |

Continued

**Table 1** Continued

| Author | Year | Title | Participants | Outcome measure |
|---|---|---|---|---|
| Matsumura *et al*[11] | 2018 | Simulating clinical psychiatry for medical students: a comprehensive clinic simulator with virtual patients and an electronic medical record system | 79 Y5 MS | Knowledge (non-validated) |
| McCaffrey *et al*[41] | 2013 | Interprofessional education in community-based Alzheimer's disease diagnosis and treatment | 110 MS and nurse practitioner trainees | Knowledge (non-validated) Attitudes (non-validated) |
| Morhardt[16] | 2013 | The effects of an experiential learning and mentorship program pairing medical students and persons with cognitive impairment: a qualitative content analysis | 71 Y1 MS | Qualitative (reflections) |
| Van Zuilen *et al*[14] | 2018 | Achieving medical student mastery in screening for cognitive impairment: results from a blended learning curriculum | 60 MS, (Y unknown) | Knowledge (non-validated) |
| Yang *et al*[42] | 2016 | 3D (dementia, delirium, depression) didactic for medical students | 23 Y3/4 MS | Knowledge (non-validated) |

Papers identified through manual searches in bold.
Partnering in Alzheimer's Instruction Research Study (PAIRS).
AADS, Adolescent Attitude to Dementia Scale; ADKS, Alzheimer's Disease Knowledge Scale; ADQ, Approaches to Dementia Questionnaire; ATOP, attitudes towards older people; BPDKT, Buddy Programme Dementia Knowledge Test; BUPPDKT, Boston University PAIRS Programme Knowledge Test; DAS, Dementia Attitudes Scale; DKAT, Dementia Knowledge Assessment Survey; DKQ, Dementia Knowledge Questionnaire; FG, focus groups; IPQ, Illness Perception Questionnaire; JSE, Jefferson Scale of Empathy; MCRS, Medical Condition Regard Scale; MS, medical student; Y, year.

*synthesis*'.[21] We synthesised the findings from our exploratory literature search with our own knowledge and experience of designing and delivering medical education about dementia to produce the following IPT illustrating the contexts where teaching and learning takes place, the mechanisms (barriers and facilitators to integration into the curriculum) and the desired outcomes, often termed CMOs (figure 2).

We condensed and structured our initial findings into four contextual barriers to implementation of effective

medical education: culture, concern for patient welfare, student attitudes to dementia and logistics. 'Culture' concerns the rigidity of curricular structures and the perceived lower status of conditions such as dementia in the hierarchy of 'ologies'. 'Concern for patient welfare' incorporates legitimate concerns about the potential vulnerability of PWD and their carers to be burdened by involvement in education. 'Student attitudes' acknowledges the perception that clinical experience of acute care is a more valuable learning experience than

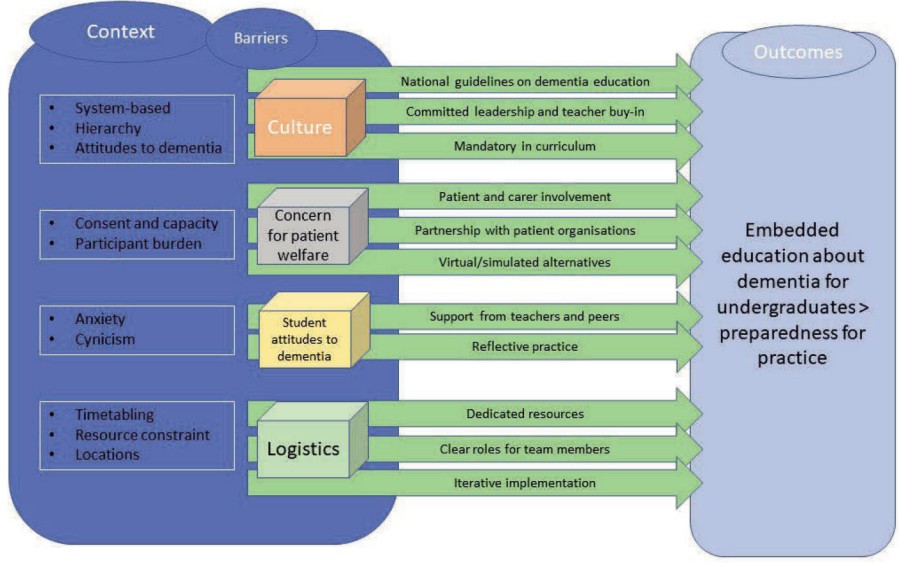

**Figure 2** Initial programme theory.

community-based engagement with PWD. 'Logistics' highlight the resource implications in terms of time and money to implement longitudinal interventions for large numbers of students. Each of the four contextual barriers are accompanied by mechanisms that facilitate surmounting these barriers in order to achieve the outcome of embedded education which prepares medical students for graduation to practice.

## DISCUSSION

Our IPT proposes four categories of barriers to implementation of effective dementia education which derive from the medical school context: culture, concern for patient welfare, student attitudes and logistics. These contextual barriers might be overcome by a range of mechanisms in order to lead to the desired outcome of embedded education that adequately prepares graduates to care for PWD. Realist synthesis encourages the development of programme theory through comparison with existing 'middle range theories'—theories available in the wider literature that are relevant to the topic but may come from close or peripheral academic disciplines.

In this case, we found that Cultural Historical Activity Theory (CHAT) held important overlap and relevance with our IPT. CHAT, or social activity theory, is of particular relevance as it is a theory that targets learning at an organisational or systems level and can be applied to help bring about educational change. Specifically, it is also relevant in circumstances where multiple voices are present. The fact that the current medical school curricula, and approaches to change, are bound by historical and deeply engrained cultural practice further highlights the relevance as typically it allows challenging of the status quo. It has been applied to dementia settings previously where innovative approaches to systems learning around dementia care have been applied. For example, in the paper by Teodorczuk et al, by defining the object as to learn about the person, its application let to innovative solutions such as relaxing visiting hours to further learn and align with the purpose of the system.[30] Moreover, multiple voices will allow engagement with the practice context and degree of dementia capability that may invariably limit implementation of any education innovations.

At play are multiple tensions that block embedding of newer dementia-focussed approaches. According to CHAT, these can be points of further expansive learning and by this manner a more fit for purpose curriculum can be developed.[31] Current divisions of labour both within education and clinical practice can be identified, and suggestions made to overcome logistical challenges to embedding more dementia focused curricula. Arguably such an approach will also lead to medical education practice becoming more patient focussed, when some have critiqued the absence of the patient in medical education.[32]

A key strength of this realist synthesis has been the flexibility to refine our study methodology in order to explain how complex educational interventions might work (or not) in particular contexts. We believe that this approach will lead to a more pragmatic answer for educators interested in improving medical education about dementia in their own institutions than a traditional systematic review. At this stage, our IPT has not yet been reviewed and refined by stakeholders outside of our study team—this limitation will be addressed in the next stage of our project. We will also address the longitudinal viability of integrating dementia education into the undergraduate curriculum, which remains unanswered at this stage.

We propose to use our IPT as the basis for ongoing realist synthesis—we will further interrogate the literature around medical education on dementia to evaluate the fit of our model and make refinements as required. We will test the refined programme theory using case studies of successful and unsuccessful implementation of educational interventions into medical school curricula, including neglected areas of teaching that may face similar factors such as learning disability. We will seek external validation of the programme theory through consultation with educators with experience of successfully integrating educational interventions about dementia into medical curricula, in addition to PWD and their carers. This will also aid our understanding of the longitudinal viability of integrating dementia education—published articles usually outline the immediate outcomes of an intervention, but seldom contain information about the longer-term impact of curriculum reform. Discussions with medical educators who have spent years in the field will help to shed light on a process that can be slow to evolve.

### Conclusion

Our IPT provides a firm basis for moving forward with our understanding of effective implementation of dementia teaching into medical education. We acknowledge that the integration of medical education about dementia is likely to be challenging in the context of established medical culture, but the changing demographics of our population and the needs of PWD warrants social accountability to do so. Realist methodology, including the application of systems change models such as CHAT, will help us to better understand the way that contextual barriers to effective medical education about dementia can be overcome.

**Author affiliations**
[1]Sunderland Medical School, University of Sunderland, Sunderland, UK
[2]School of Dental Sciences, Newcastle University, Newcastle upon Tyne, UK
[3]North East and North Cumbria GP Training Programme, Health Education England, Newcastle upon Tyne, UK
[4]Graduate School of Medicine, University of Wollongong, Wollongong, New South Wales, Australia
[5]School of Medicine and Dentistry, Griffith University, Southport, Queensland, Australia
[6]Northside Clinical Unit, TheUniversity of Queensland, Brisbane, Queensland, Australia

⁷Older Peoples Mental Health, Metro North Mental Health, The Prince Charles Hospital, Brisbane, Queensland, Australia
⁸School of Nursing, Queensland University of Technology, Kelvin Grove, Brisbane, Queensland, Australia

**Contributors** ET conceptualised the idea, approach, constructed the research team and is the guarantor. ET, LW, RP, TKK and AT provided feedback and further developed the idea and realist synthesis approach. RP was responsible for the primary search of initial studies. ET, LW, RP, TKK and AT contributed to the search refinement, data extraction and analysis. ET summarised findings and drafted an initial programme theory to which LW, RP, TKK and AT provided feedback and further developed. ET prepared the initial draft manuscript and LW, RP, TKK and AT reviewed and developed the manuscript. All authors (ET, LW, RP, TKK and AT) accepted the final version.

**Funding** The authors have not declared a specific grant for this research from any funding agency in the public, commercial or not-for-profit sectors.

**Competing interests** None declared.

**Patient and public involvement** Patients and/or the public were not involved in the design, or conduct, or reporting, or dissemination plans of this research.

**Patient consent for publication** Not applicable.

**Ethics approval** Not applicable.

**Provenance and peer review** Not commissioned; externally peer reviewed.

**Data availability statement** No data are available.

**ORCID iD**
Luisa Wakeling http://orcid.org/0000-0002-0292-3680

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
