## [Reviewer comments · BMJ Open]

ARTICLE DETAILS

TITLE (PROVISIONAL)	Lost in translation: How can education about dementia be effectively integrated into medical school contexts? A realist synthesis
AUTHORS	Tullo, Ellen; Wakeling, Luisa; Pearse, Rachel; Kheng Khoo, Tien; Teodorczuk, Andrew

VERSION 1 – REVIEW

REVIEWER	Soo Borson University of Washington
REVIEW RETURNED	09-Jul-2023

GENERAL COMMENTS	This is a welcome contribution that updates the literature on medical student learning about dementia using a pragmatic approach to review that makes it particularly useful. The concept and importance of the work are clearly articulated and the manuscript is well written. The IPT is itself a conceptual contribution that a wide range of readers can benefit from. In addition to these highly positive contributions, there is, to my mind, a missing piece: that's the fact that medical students are in training to be doctors in actual health care systems, and most health care systems are in early stages of evolving toward "dementia-capability" - if they are in any stage at all. Recognizing that health care delivery works differently in different countries, that social, cultural, and financing differences are important influences on these differences, and that conditions requiring complex multi domain care - like dementia - are more vulnerable to a sluggish pace of improvement than those that require discrete (e.g. surgical or pharmacological) interventions. Nevertheless, I would like to see you link your work to the practice context in which medical students will ultimately work as doctors; this is an unacknowledged but likely a potent driver of why it's hard to change education.
--

REVIEWER	Ani Orchanian-Cheff University Health Network, Library and Information Services
REVIEW RETURNED	21-Jul-2023

GENERAL COMMENTS	Thank you for the privilege of reviewing your realist synthesis on education about dementia in medical schools. My primary concerns are about the search reporting. It is not sufficiently transparent and reproducible. I would suggest that you might consider following PRISMA for searching as a good guide http://www.prisma-statement.org/Extensions/Searching. 1. For the databases searched, please include the vendors. For example, was Embase searched through Embase.com or Ovid? Was CINAHL searched through Ebsco? What vendor did you
---

	access Psycinfo from? 2. Please also indicate the date the search was completed. To be current this search should be less than a year old. If it isn't, I would update the search to capture newer results. 3. For each of the databases searched, copy and paste the exact search strategy into an appendix. This is what is needed to be able to reproduce your search. This is also in keeping with the RAMSES guidelines "Provide details on all the sources accessed for information in the review. Where searching in electronic databases has taken place, the details should include, for example, name of database, search terms, dates of coverage and date last searched." 4. For the flow diagram of your results, in the first box it is also helpful to know how many results you obtained from each source searched. This in addition to n. 5. The search, in order to be systematic and comprehensive, should include a combination of subject headings and key words for all concepts searched. I suspect that is why your search was not sufficiently broad to capture all citations. As such, in addition to using MeSH terms (in your PubMed search) you would need to include key words for the concepts of medical education, medical schools and neurocognitive disorders/dementia. This practice should also be employed in all other databases, but their subject headings would be different as they do not use MeSH. 6. Citation 9 is missing a date.
--	--

REVIEWER	Stephanie Daley Brighton and Sussex Medical School, Centre for Dementia Studies
REVIEW RETURNED	31-Jul-2023

GENERAL COMMENTS	This is a very well-written and relevant paper. I have only three points Whilst I agree with the themes identified, I would have like to have seen more discussion/exploration as to some of them - for example; concern as about patient welfare as a possible barrier, and whether in fact this espoused concern actually holds up to scrutiny given the number of evidence based programmes which do directly involve people with dementia as co-educators without apparent harm. 2. It would have been useful to explore whether there are similar factors in other important, but neglected areas of teaching, for example, intellectual disabilities, and whether there are lessons to be learnt from these other areas. The Oliver McGowan Mandatory Training in Learning Disability and Autism is an example where a bereaved family member has brought both political and policy change to medical schools (and healthcare providers) - are there lessons learnt. 3) Linked to the above point, what is the impact of a lack of coherent current dementia policy agenda in relation to dementia.
---

VERSION 1 – AUTHOR RESPONSE

Reviewer: 1

Prof. Soo Borson, University of Washington

Comments to the Author:

This is a welcome contribution that updates the literature on medical student learning about dementia using a pragmatic approach to review that makes it particularly useful. The concept and importance of the work are clearly articulated and the manuscript is well written. The IPT is itself a conceptual

contribution that a wide range of readers can benefit from.

In addition to these highly positive contributions, there is, to my mind, a missing piece: that's the fact that medical students are in training to be doctors in actual health care systems, and most health care systems are in early stages of evolving toward "dementia-capability" - if they are in any stage at all. Recognizing that health care delivery works differently in different countries, that social, cultural, and financing differences are important influences on these differences, and that conditions requiring complex multi domain care - like dementia - are more vulnerable to a sluggish pace of improvement than those that require discrete (e.g. surgical or pharmacological) interventions. Nevertheless, I would like to see you link your work to the practice context in which medical students will ultimately work as doctors; this is an unacknowledged but likely a potent driver of why it's hard to change education.

This is an excellent point. We have included reference to this within the expanded introduction and raise the issues of the hidden curriculum and also sub optimal systems of practice. It also further supports the choice of CHAT as having relevance within the programme theory and as such we refer to the importance of practice context in the discussion as well.

Reviewer: 2

Ms. Ani Orchanian-Cheff, University Health Network

Comments to the Author:

Thank you for the privilege of reviewing your realist synthesis on education about dementia in medical schools. My primary concerns are about the search reporting. It is not sufficiently transparent and reproducible. I would suggest that you might consider following PRISMA for searching as a good guide <http://www.prisma-statement.org/Extensions/Searching>.

1. For the databases searched, please include the vendors. For example, was Embase searched through Embase.com or Ovid? Was CINAHL searched through Ebsco? What vendor did you access Psychinfo from?

Now included in supplementary file

2. Please also indicate the date the search was completed. To be current this search should be less than a year old. If it isn't, I would update the search to capture newer results.

The most recent list of references were derived from searches in April 2021. Updating the search in August 2023 highlighted four additional eligible papers, two of which we have already identified via snowball/manual searching. We will review all four in the refinement stage of our project, alongside the expanded scope to parallel fields (please see comments above)

3. For each of the databases searched, copy and paste the exact search strategy into an appendix. This is what is needed to be able to reproduce your search. This is also in keeping with the RAMSES guidelines "Provide details on all the sources accessed for information in the review. Where searching in electronic databases has taken place, the details should include, for example, name of database, search terms, dates of coverage and date last searched."

Now included in supplementary file

4. For the flow diagram of your results, in the first box it is also helpful to know how many results you obtained from each source searched. This in addition to n.

Now included in supplementary file

5. The search, in order to be systematic and comprehensive, should include a combination of subject headings and key words for all concepts searched. I suspect that is why your search was not

sufficiently broad to capture all citations. As such, in addition to using MeSH terms (in your PubMed search) you would need to include key words for the concepts of medical education, medical schools and neurocognitive disorders/dementia. This practice should also be employed in all other databases, but their subject headings would be different as they do not use MeSH.

Now included in supplementary file

6. Citation 9 is missing a date.

Now added

Reviewer: 3

Dr. Stephanie Daley, Brighton and Sussex Medical School

Comments to the Author:

This is a very well-written and relevant paper. I have only three points

1. Whilst I agree with the themes identified, I would have like to have seen more discussion/exploration as to some of them - for example; concern as about patient welfare as a possible barrier, and whether in fact this espoused concern actually holds up to scrutiny given the number of evidence based programmes which do directly involve people with dementia as co-educators without apparent harm.

2. It would have been useful to explore whether there are similar factors in other important, but neglected areas of teaching, for example, intellectual disabilities, and whether there are lessons to be learnt from these other areas. The Oliver McGowan Mandatory Training in Learning Disability and Autism is an example where a bereaved family member has brought both political and policy change to medical schools (and healthcare providers) - are there lessons learnt.

3. Linked to the above point, what is the impact of a lack of coherent current dementia policy agenda in relation to dementia.

Thank-you for these 3 comments. Each of these issues will be addressed in the refinement stage of this project when we return to the evidence base and reach out to stakeholders (educators, students and patients) for their views and experiences of delivering dementia education. Extension into other fields of teaching (such as intellectual disabilities) is likely to add helpful messages which can be incorporated into our final model and this is now acknowledged in the discussion.

VERSION 2 – REVIEW

REVIEWER	Soo Borson University of Washington
REVIEW RETURNED	23-Sep-2023

GENERAL COMMENTS	This version is responsive to previous critiques and makes a significant contribution to the literature. Dementia is seriously underrepresented in undergraduate medical training and is necessary to prepare the workforce of tomorrow. Your realist review and model are adaptable to various data sets and will help to advance the field.
---

REVIEWER	Ani Orchanian-Cheff University Health Network, Library and Information Services
REVIEW RETURNED	02-Oct-2023

GENERAL COMMENTS	Thank you for revising your manuscript. I still have concerns about the search strategy. For a comprehensive search you need a combination of subject headings and keywords for your main concepts. In your search you have only used subject headings. That is why your results are so low. For the topic of dementia you can get a search strategy from Cochrane https://www.cochranelibrary.com/cdsr/doi/10.1002/14651858.CD013572.pub2/appendices#CD013572-sec-0070 that you can adapt to pubmed. At the very least you would need to use dementia as a keyword search. As well, if you do not supplement your search strategy with keywords you will miss on citations that are not yet indexed. Please expand your searches to include keywords for each of your concepts. For the Embase search you indicate that you used Ovid. The search as presented is not in the syntax of Ovid. Did you copy and paste the strategy exactly as run? Please include the names of the database vendors in your actual manuscript as well.
---

VERSION 2 – AUTHOR RESPONSE

Reviewer 1 and thank them for highlighting the significant contribution of our paper.

Reviewer 2 again highlights concerns about the exact terminology and syntax of our original searches. We would like to reiterate that this is a realist approach that accepts an initial process of exploratory scoping of the literature, rather than fixed systematic criteria, in order to develop an initial programme theory. This is the first step in our process of realist synthesis – proposed ongoing realist methodology to iteratively test and refine the model through ongoing literature searching is outlined in the discussion.

We are grateful for the suggestions as to comprehensive means of including all possible search terms for the topic of dementia for different databases and will utilise these strategies in our ongoing literature searches for the testing of our programme theory. However, it is not practical at this point to re-run the initial searches and is unlikely to change the key concepts included in our programme theory. To fulfil this would be to adopt a literature search method that is not in keeping with a realist approach and we ask that this request is reconsidered.